# The Joint Effect of Grazing Intensity and Soil Factors on Aboveground Net Primary Production in Hulunber Grasslands Meadow Steppe

**Ahmed Ibrahim Ahmed [1,2]**, **Lulu Hou [1]**, **Ruirui Yan [1]**, **Xiaoping Xin [1,\*]** and **Yousif Mohamed Zainelabdeen [1,2]**

[1] Laboratory of Grassland Science, Institute of Agricultural Resources and Regional Planning, Chinese Academy of Agricultural Sciences, No 12, Zhongguancun South Street, Haidian District, Beijing 100081, China; abuaawab@gmail.com (A.I.A.); 82101176057@caas.cn (L.H.); yanruirui@caas.cn (R.Y.); yousifmohamed80@yahoo.com (Y.M.Z.)

[2] Agricultural Research Corporation (ARC), Wad Medani 126, Sudan

\* Correspondence: xinxiaoping@caas.cn

**Abstract:** The management practices required for grazing management will continue to increase, as necessitated by the reported rate of reduction in productivity, coupled with the degradation of Inner Mongolian steppe ecosystems. The current study was conducted to (i) examine the responses of aboveground net primary production (ANPP) to different grazing intensities and its relationship with soil factors and (ii) study the effects of grazing intensity on herbage growth and dry matter intake in Hulunber grasslands, Northeastern China. Six grazing rate treatments (G0.00, G0.23, G0.34, G0.46, G0.69, and G0.92 animal unit (AU ha$^{-1}$) for zero, two, three, four, six, and eight young cows with ranging weight of 250–300 kg/plot), with three replications, were established during two consecutive growing seasons in 2017 and 2018. Our study concentrated on the grazing-induced degradation processes by different intensities of grazing. The highest decrease in aboveground biomass (AGB) was 64.1% and 59.3%, in 2017 and 2018, respectively, by the G0.92 treatment as compared with the G0.00 treatment. There was a positive relationship between yearly precipitation and ANPP. The grazing tolerance and growth rate of forage were higher in the wet year than in the dry year. Understanding the ecological consequences of grazing intensity provides useful information for assessing current grazing management scenarios and taking timely adaptation measures to maintain grassland capacity in a short and long-term system.

**Keywords:** aboveground biomass; herbage growth; aboveground net primary production; grazing experiment; grazing intensity

## 1. Introduction

The Hulunber grasslands area forms a portion of the Inner Mongolian typical steppe ecosystem located in Northeastern China. It is one of the biggest areas of natural temperate sub-humid prairie grasslands in the world, with an estimated coverage area of $9.97 \times 10^6$ km$^2$. Though a considerable portion of the total area has been transformed into cultivated lands in the past five decades, a huge part of the region is still used as natural or semi-natural grasslands for seasonal or constant grazing by cattle and sheep, causing 50% of obtainable grassland degradation in the area [1–4]. For now, suitable stocking rate limitation is one of the most critical and insistent means being adopted to support the grasslands adaptive management and sustainable development [5,6]. Due to the abundance of plant species with high palatability, the typical steppe has become quite suitable for making hay and grazing activities [7–9]. Due to the high grazing potential of the area [10], livestock husbandry controlled by

small ruminants (mainly sheep and goats) has resulted in serious degradation, which can basically be demonstrated by the reduced state of canopy cover [11]. Therefore, in recent decades, the typical steppe ecosystem size and productivity has declined drastically [7,12]. The reason for this phenomenon is that, in recent decades, the shift from seminomadic to settled animal husbandry has led to the change of concentration and land use.

Rainfall is always considered as the key factor to control the productivity of semiarid grassland and determine the suitable grazing rate [11,13–16]. In grassland ecosystems, climate is the main driving factor for plant growth and species dynamics [17]. It is reported that the inter-annual variation of precipitation and temperature is closely related to the aboveground net primary productivity and vegetation dynamics (such as plant composition and species diversity) [18,19]. However, the analysis of grassland vegetation dynamics is mainly from the perspective of annual average precipitation, and the impact of precipitation or temperature change in the year is not clear. In addition to climate and environmental conditions, management strategies also affect grazing intensity. The time and intensity of grazing are the most important controllable factors affecting the response of plants to a grazing [20]. Therefore, in order to optimize the stocking rate and yield of the Inner Mongolia grassland ecosystem, alternative management measures should be taken. The multiple grazing effects of long-term grazing on typical grasslands have been studied [7,10,21,22]. Considering the grazing history, these studies found the influence of long-term grazing on pasture and grassland structure, which was confused with the qualitative evaluation of grazing intensity [15,23]. A few experiments were carried out on typical grassland, to analyze the response of plants under controlled grazing rate.

Previous studies reported that the decline of aboveground biomass, vegetation coverage reduction, and increased soil water evaporation can be attributed to grazing intensity [24,25], erosion of soil due to wind [26], and soil nutrients and litter decompositions [25]. These grazing effects were confirmed by several studies and demonstrate increases in soil C and N, with an increase in aboveground biomass and ground cover following the exclusion of grazing [27,28]. Moreover, C inputs decrease from roots to soil due to grazing [29]. Consequently, this leads to decreases in soil organic carbon (SOC) and total nitrogen (TN) with an increase in grazing intensity. In other terms, the higher the production of biomass, the higher the soil content of SOC and N. This can be attributed probably to the fact that the main source and pool of soil C and N is the soil organic matter [30]. Therefore, lower production of biomass and lower content of N and C can be anticipated due to intensive grazing.

Inner Mongolia lacks detailed information on the degradation process and its formation. Therefore, the current study was carried out, firstly, to examine the effects of grazing on the aboveground net primary production (ANPP) and, secondly, to clarify the potential response of ANPP to animal intake variables in Hulunber area (Northeast of Inner Mongolia), in order to find an alternative conventional livestock husbandry management pattern. Under six grazing intensities, ranging from non-grazing to heavy grazing, a controlled grazing experiment on the influence of cattle grazing on vegetation parameters was established. This wide range of stocking rates may require extensive analysis of vegetation cover response to grazing [31,32]. Therefore, the current study explains whether the increase of ANPP caused by grazing is based on the assumption of grazing optimization [32–35]. Moreover, the aboveground biomass inside and outside the cages is evaluated, to better understand the effect of animal intake on the aboveground net primary production. We needed to conduct a measurable experiment to investigate the impact of grazing on the aboveground net primary production of Hulunber grassland, in which grazing compression and topography were variables. The human population is growing with a corresponding increased demand for animal-derived foods. This consequently calls for intensified livestock production. However, associated net effects act on semiarid grasslands through overgrazing and desertification. Strategies for sustainable livestock production that conserves vegetation and fulfills farmers' economic interests are highly needed. The aboveground biomass measurements for more confirmation are applied.

## 2. Materials and Methods

### 2.1. Study Area

The present study was conducted at the Hulunber Grassland Ecosystem Observation and Research Station located at Xiertala Farm, in the center of the Hulunber meadow steppe (49°19′349″ N, 119°56′521″ E) in the northeastern region of Inner Mongolia, China (Figure 1). The elevation varies from 666 to 680 m. The climate is characterized as continental, temperate, and semiarid, with an annual average of 110 frost-free days. The annual mean rainfall ranges from 350 to 400 mm, approximately 80% of which falls between July and September. The annual mean air temperature in this area is −5 to −2 °C, with the highest monthly mean of 36.2 °C in July and the lowest of −48.5 °C in January. The vegetation cover is characterized as a typical *Leymus chinensis* and forbs meadow steppe. The dominant species are *L. chinensis*, *Scutellaria baicalensis*, *Carex pediformis*, *Galium verum*, *Bupleurum scorzonerifolium*, and *Filifolium sibiricum*. The soil is characterized as a chernozem, or chestnut soil. Active plant growth occurs between June and October, when temperatures rise above 15 °C.

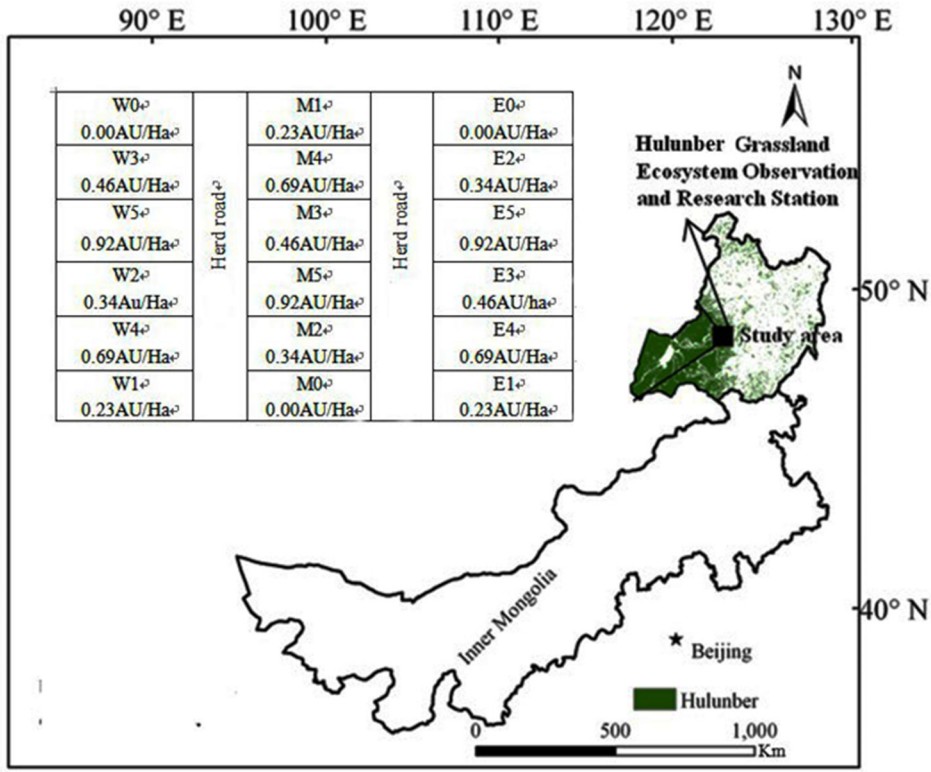

**Figure 1.** Study area geographic location and the experiment layout.

### 2.2. Experimental Design and Fieldwork

The grazing experiment was designed with six grazing intensities (0.00, 0.23, 0.34, 0.46, 0.69, and 0.92 Animal Units ha$^{-1}$, where 1 Animal Unit (AU) = 500 kg of adult cows). Three replicates were used for each stocking rate, with each replicated area covered a 5-hectare paddock. A total of 18 plots were randomly distributed over a total homogeneous area of 90 ha. The grazing intensities were achieved by using 0, 2, 3, 4, 6, or 8 young cows with body weight ranging from 250 to 300 kg per plot. Constant grazing was held for 120 days, from June to September, during the 2017 and 2018 growing seasons. The cows were preserved day and night in the grazing plots, and an outside water source was used to supply them with drinking water. The field work was carried out at the growing seasons of 2017 and 2018. Measurements were made three times per month in 2017, and four times per month in 2018, excluding September, where 3 measurements were taken due to prevailing weather conditions.

In addition, during the growing season, from June to September in 2017 and 2018, six quadrat samples of $50 \times 50$ cm were placed randomly in each grazing area inside and outside the cages.

### 2.3. Forage Dry Matter Intake (DMI) Estimated Using Grazing Cages (Cage Technique)

To estimate forage disappearance before and after grazing, the grazing cage technique [36] was used. In each plot, six cages of approximately $50 \times 50$ cm were used during the grazing period. Forage within and outside the cages were clipped from $0.25$ m$^2$ quadrats every ten days of grazing. The plant species in each quadrat were clipped at 2.5 cm above ground level, the harvested biomass species were placed in paper bags and were oven-dried for 48 h at 65 °C, to constant weight [37]. The dried materials were weighed by using a sensitive balance. The amount of forage consumed by grazing animals was estimated as the result of the difference between un-grazed cages (inside) and grazed cages (outside). The dry matter intake was calculated by using the following equation:

$$\text{DMI (kg/d)} = \frac{[\text{DMI inside cage (kg/ha)} - \text{DMI outside cage (kg/ha)}] \times \text{area (ha)}}{\text{Number of grazing days}} \tag{1}$$

### 2.4. Grazing Intensity and ANPP

The variance between the consumed biomass by cattle and the obtainable biomass was defined as grazing intensity. Available biomass was considered as the sum or total of the ANPP at the beginning of the grazing season and the biomass. The consuming of ANPP and biomass was calculated by the moving cages method [38,39]. The sum of increments in the aboveground biomass sample was calculated in 2017 every 10 days, from June to the end of September, and in 2018, every 7 days, to calculate the ANPP. Plant biomass was sampled inside and outside the moveable cages within 6 randomly located $50 \times 50$ cm quadrats, and then dried out at 65 °C for 48 h, to achieve the fixed weight. In 2017 and 2018, cattle consumption was estimated by aggregating differences in vegetation biomass collected inside and outside the movable cages for 10 and 7 days, respectively. The following equation was used to evaluate the ANPP:

$$\text{ANPP} = W_{1g} + (W_{2u} - W_{1g}) + (W_{3u} - W_{2g}) \tag{2}$$

where $W_i$ represents the dry matter weight of standing biomass at sample time $t_i$ (i = 1, 2, 3, 4: every 10 days in 2017 and every 7 days in 2018 in (June, July, August, and September, respectively). Indices *u* and *g* for un-grazed and grazed, respectively, represent samples taken within and outside the exclosure cages.

### 2.5. Growth Rate (GR) and Relative Growth Rate (RGR)

The seasonal averages [40] of the control plot and the hay production area were defined as the absolute growth rate (GR) and the relative growth rate (RGR), and they were calculated by the following equations:

$$\text{GR} = (W_{2u} - W_{1g}) / (t_2 - t_1) \tag{3}$$

$$\text{RGR} = [\ln (W_{2u}) - \ln(w_{1g})] / t_2 - t_1) \tag{4}$$

where $W_i$ represents the ANPP dry matter weight of the inside cage at sampling time ti (i = 1, 2) inside (*u* indicates un-grazed) and outside (*g* indicates grazed) the exclosure cages.

### 2.6. Soil Moisture and Temperature Measurements

ECH$_2$O 5TE sensors (decagon equipment, Pullman, WA, USA) were used to estimate soil moisture content (cm$^3$ cm$^{-3}$) and soil temperature (°C) [41–43]. 5TE sensors were calibrated before fixing, and the soil temperature and humidity were recorded once every 10 s, and once every 10 min, and averaged

for final storage. In the 12 experimental field plots, we established a 1-meter soil sampling area and used 60 sensors to measure soil moisture and soil temperature at 10, 20, 40, 60, and 100 cm depths.

### 2.7. C3 and C4 Plants

In grassland grass fields, the plant compositions are predominantly C3 and C4 plants. C3 and C4 grassland grasses varied physiologically and morphologically. Besides C3's grass sensitivity to low temperatures and the preference of C4 grass to the warm or hot weather conditions, C3 grasses are also found to be more nutritious and palatable for pastoral livestock. Grazing by amending species composition and competition within the species predominantly promotes the physiological and morphological variations between C3 and C4 grasses.

### 2.8. Soil Nutrient Contents

In early August of 2017 and 2018, soil samples (0–10 cm) were collected from 10 points in each plot, for analysis of soil nutrient contents [44]. The soil particle size distribution was measured by a Mastersizer 2000 laser particle size analyzer (0–2000 μm, Malvern Panalytical Ltd, Malvern, UK), the soil organic carbon content (SOC) was measured by using the dichromate oxidation method, the total nitrogen content (TN) was measured by using the semi-micro Kjeldahl method, and the total amount of phosphorus (TP) was estimated by using the molybdenum antimony resistance colorimetric method. SOC, TN, and TP were calculated by using the following equations:

$$\text{SOC}\left(g/kg^{-1}\right) = \frac{\frac{c \times 5}{V_0} \times (V_0 - V) \times 10^{-3} \times 3.0 \times 1.1}{m \times k} \times 1000 \tag{5}$$

where $c$ = 0.8000 mol·L$^{-1}$ (1/6 $K_2Cr_2O_7$) the concentration of the standard solution, 5 = volume added to standard solution of potassium dichromate (mL), $V_0$ = the blank titration uses the de-FeSO$_4$ volume (mL), $V$ = the sample was titrated with FeSO$_4$ volume (mL) 1000, 3.0 = 1/4 molar mass of a carbon atom (g·mol$^{-1}$), $10^{-3}$ = convert mL to L, 1.1 = oxidation correction factor, $m$ = air-dried soil sample quality (g), and $k$ = the air-dried soil sample converted into the coefficient of drying soil.

$$\text{TN}\left(g/kg^{-1}\right) = \frac{(V - V_0) \times c(\frac{1}{2}H_2SO_4) \times 14.0 \times 10^{-3}}{m} \times 10^3 \tag{6}$$

where $V$ = the volume of the standard solution of acid used in the titrate (mL), $V_0$ = the volume of the acid standard solution used when titrating blank (mL), $c$ = 0.01 mol·L$^{-1}$ (1/2 $H_2SO_4$) or HCl standard solution concentration, 14.0 = the molar mass of the nitrogen atom (g·mol$^{-1}$), $10^{-3}$ = convert mL to L, and $m$ = quality of dried soil samples (g).

$$\text{TP}\left(g/kg^{-1}\right) = \rho \times \frac{V}{m} \times \frac{V_2}{V_1} \times 10^{-3} \tag{7}$$

where $\rho$ = the mass concentration of phosphorus in the solution to be measured (g/kg$^{-1}$), $V$ = the number of mL of the sample preparation solution, $m$ = dried soil quality (g), $V_1$ = extract filtrate mL, $V_2$ = volume of colored solution (mL), and $10^{-3}$ = Convert μg to g·kg$^{-1}$.

### 2.9. Statistical Analysis

The data analysis was carried out by using SAS Statistics Software Program (version 9.2, SAS Institute, Cary, NC, USA) [45] to examine the significant differences in ANPP and dry matter intake. One-way ANOVA was used under the various grazing intensities in the same years. A two-way ANOVA was used to test the effects of grazing, year, and their interactions for AGB, ANPP, and the herbage growth rate for both 2017and 2018 seasons. Principal component analysis (PCA), using the software CANOCO 4.5 [46], was used to ordinate the grassland vegetation and environmental variables.

## 3. Results

### 3.1. The Relationship of Climatic Conditions with Productivity Levels

In 2017, the annual precipitation of 210.9 mm was lower than the long-term average, while in 2018, the annual precipitation recorded 349.1 mm, which was almost similar to the long-term average. In 2017, the highest rates of rainfall were recorded in July and August, whereas in 2018, the highest rainfall amounts have been registered in June and July. In 2017, a drought was observed, and the average precipitation in May to September was 189.2 mm, with an 84.3 mm peak in July, while in 2018, the average rainfall in May to September was 342.2 mm, and the peak was recorded in July, with 117.2 mm (Figure 2). Generally, 2018 recorded the highest AGB, ANPP, and herbage growth rate.

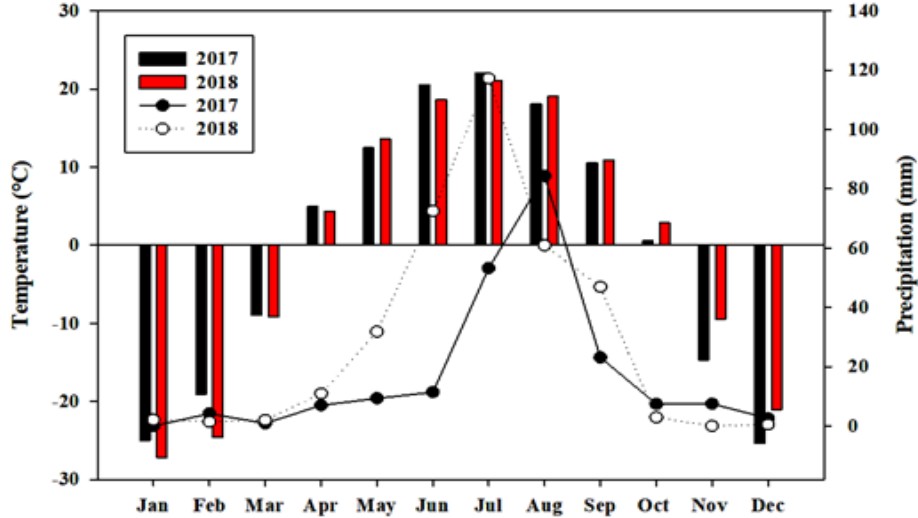

**Figure 2.** Monthly average temperature and precipitation over selected periods, at the experimental site in the meadow steppe of Hulunber, Inner Mongolia.

### 3.2. Grazing Effects on Aboveground Biomass (AGB)

The AGB was higher in 2018 than in 2017, and it increased and then decreased with increasing grazing intensity. The AGB of G0.00 was higher than G0.23, but it had the same seasonal distribution in 2017 and 2018. Grazing intensity has a significant effect on AGB and showed a significant inter-annual change ($p < 0.05$, Table 1). The AGB of the G0.00 treatment was significantly higher than other treatments in each year (Figure 3). The greatest decrease in AGB was 64.1% and 59.3%, in 2017 and 2018, respectively, on the G0.92 treatment, as compared with the G0.00 treatment ($p < 0.05$).

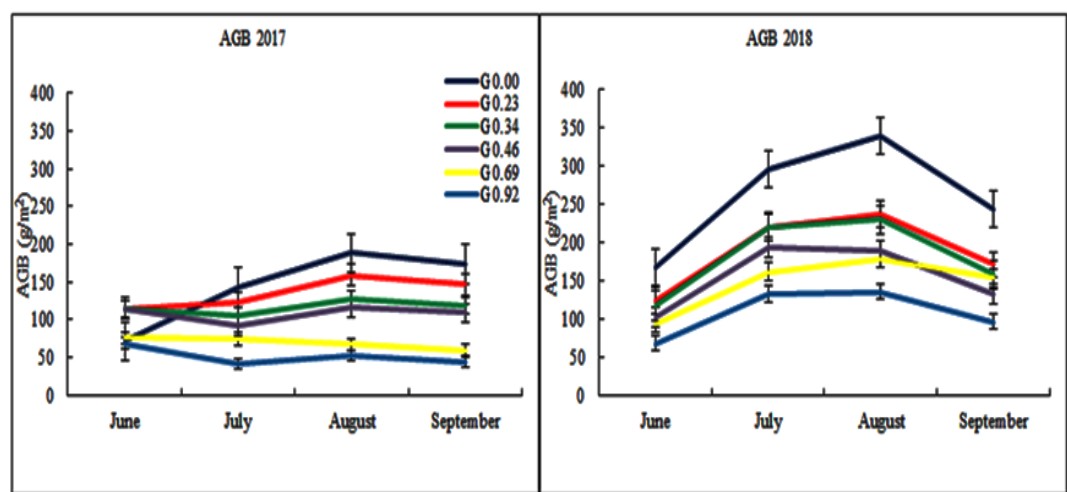

**Figure 3.** The effect of grazing intensity and season on aboveground biomass (AGB).

### 3.3. Grazing Intensity Effects on Aboveground Net Primary Production (ANPP)

Grazing intensity and annual grazing intensity had significant effects on the ANPP production unit ($p < 0.05$) (Table 1). Significant differences ($p < 0.05$) were observed among the treatments for ANPP in both 2017 and 2018. In 2017, G0.23 had the highest ANPP, which was statistically similar to G0.34. However, in 2018, the G0.00 treatment had the highest ANPP, which was statistically at par with the other treatments, except for G0.92, which had the lowest ANPP (Figure 4). ANPP decreased with increased grazing intensity and decreased significantly at G0.92. In 2017, ANPP unit production decreased by 56% and 71% at G0.92, compared with G0.00 and G0.23, respectively, and by 67% and 63% at G0.92, compared with G0.00 and G0.23, respectively (Figure 4). In 2017 and 2018, with the increase in grazing intensity, ANPP decreased significantly. Generally, in 2017 and 2018, grazing reduced the productivity of production units, e.g., G0.00 and G0.23 in 2017 and 2018 (Figure 4). Moreover, after mid-August, the ANPP decreased as a result of the end of plant growth.

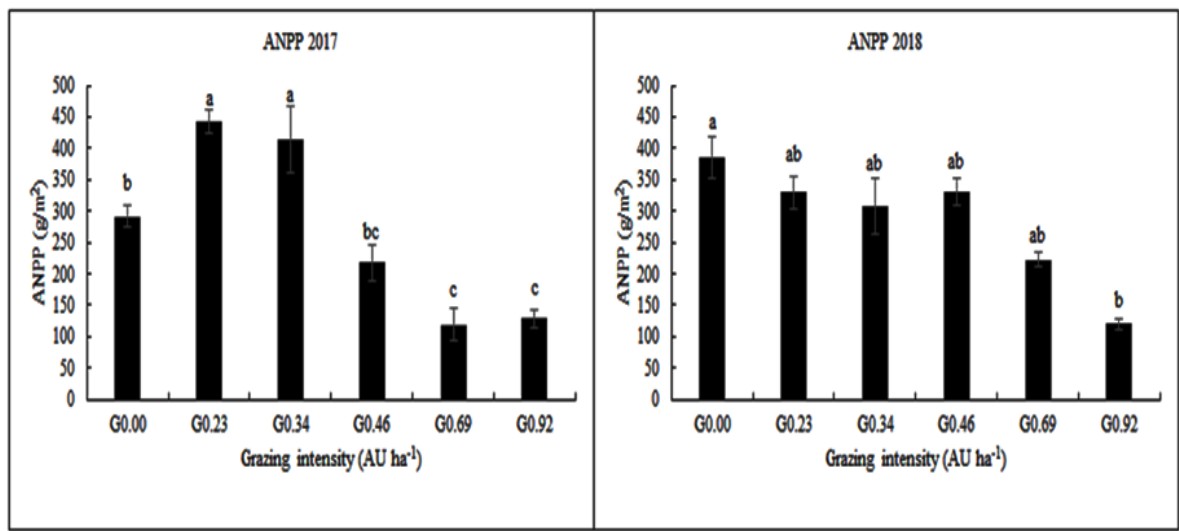

**Figure 4.** The effect of grazing intensity on aboveground net primary production at the continuous grazing system. a, b, c, means followed by the same small letters are not significantly different ($p < 0.05$).

**Table 1.** Results ($p$-values) of two-way ANOVA on the effects of grazing, year, and their interactions on aboveground biomass (AGB), aboveground net primary production (ANPP), growth rate (GR), and relative growth rate (RGR).

|  | AGB | ANPP | GR | RGR |
|---|---|---|---|---|
| Grazing | 0.000 | 0.431 | 0.089 | 0.005 |
| Year | 0.000 | 0.005 | 0.058 | 0.016 |
| Grazing * Year | 0.499 | 0.188 | 0.004 | 0.004 |

\* is the interaction between grazing and year.

### 3.4. Grazing Effects on the Dry Matter Intake

The study detected a strong negative effect of grazing intensities on DMI ($p \leq 0.05$) (Figure 5). The DMI was 14.1, 10.4, 8.3, 3.6, and 3.9 kg/d.AU for G0.23, G0.34, G0.46, G0.69, and G0.92, respectively, in 2017, while in 2018; the DMI was 24.4, 15.5, 12.3, 7.6, and 3.7 kg/d.AU for G0.23, G0.34, G0.46, G0.69, and G0.92, respectively. The DMI at G0.69 and G0.92 during the growing season were the lowest in both years and were not significantly different ($p > 0.05$) from each other. Generally, as the grazing intensities increase, the DMI decreases.

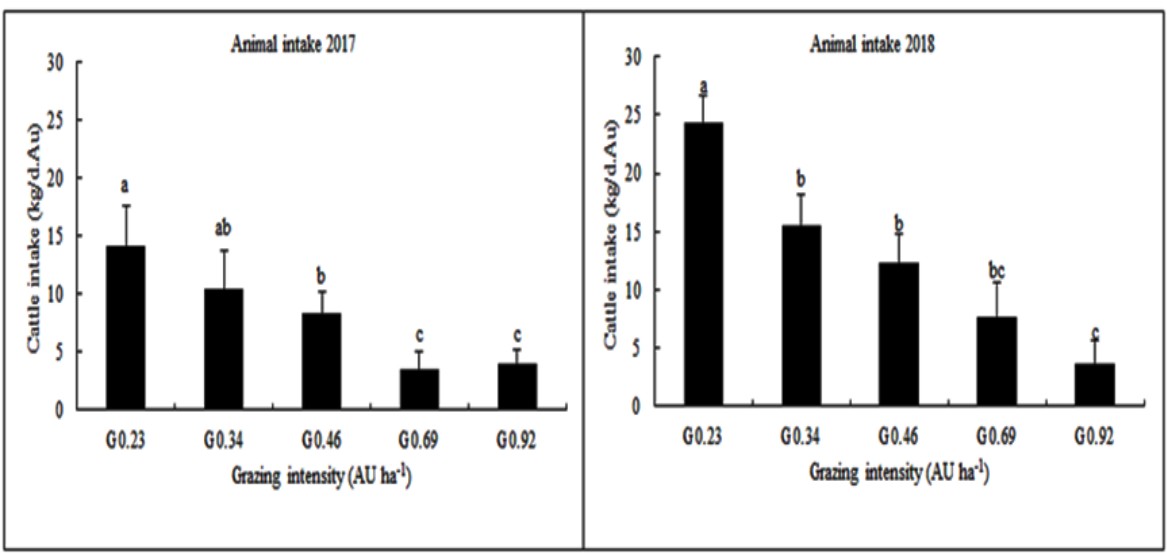

**Figure 5.** The effect of grazing intensity and season on dry matter intake (DMI). a, b, c, means followed by the same small letters are not significantly different ($p < 0.05$).

### 3.5. The Effect of Grazing Intensity on the Herbage Growth

The GR and RGR were influenced highly by sampling time and grazing intensity (Table 1). As shown in Figures 6 and 7, during August 2017, the average maximum growth rates were recorded at G0.23 and G0.34, with a GR of 6.62 and 5.68 g DM/m$^2$.day, corresponding to RGR of 0.19 and 0.18%/day, respectively. The lowest growth rates continued to decline to lower levels in September. Negative GR of −0.49 g and 0.01 g DM/m$^2$.day and RGR of −0.02%/day and −0.00%/day were observed at G0.69 and G0.92 grazing intensities, respectively. In 2018, high levels of GR were recorded in June and July, with a maximum of 5.71 g DM/m$^2$.day in July, at G0.34. In 2018, the GR decreased to a minimum of 0.41 g DM/m$^2$.day in August, at G0.34, and −6.42 g DM/m$^2$.day at G0.00 in September. In June of 2017, the GR remained at the lowest level for all the treatments, except for G0.00, which had the highest GR of 4.17 g DM/m$^2$.day. The maximum RGRs of 0.21 (G0.23) and 0.13 (G0.46) %/day were observed in July of 2017 and 2018, respectively. Among the grazing intensities, highly dynamic growth rates were detected within the different growth periods (Figure 6). The RGR was relatively constant between grazing intensities in different periods (Figure 7).

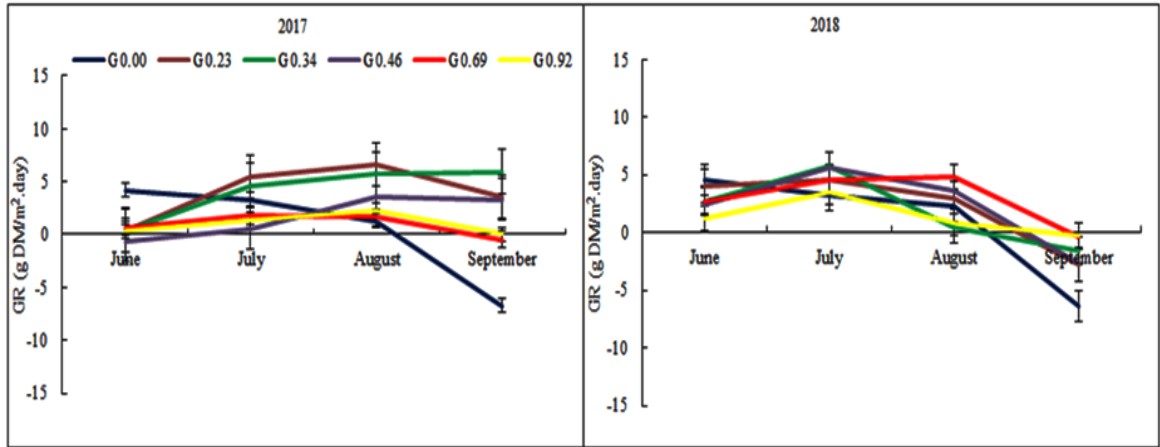

**Figure 6.** Effect of grazing intensity and season on growth rate (GR).

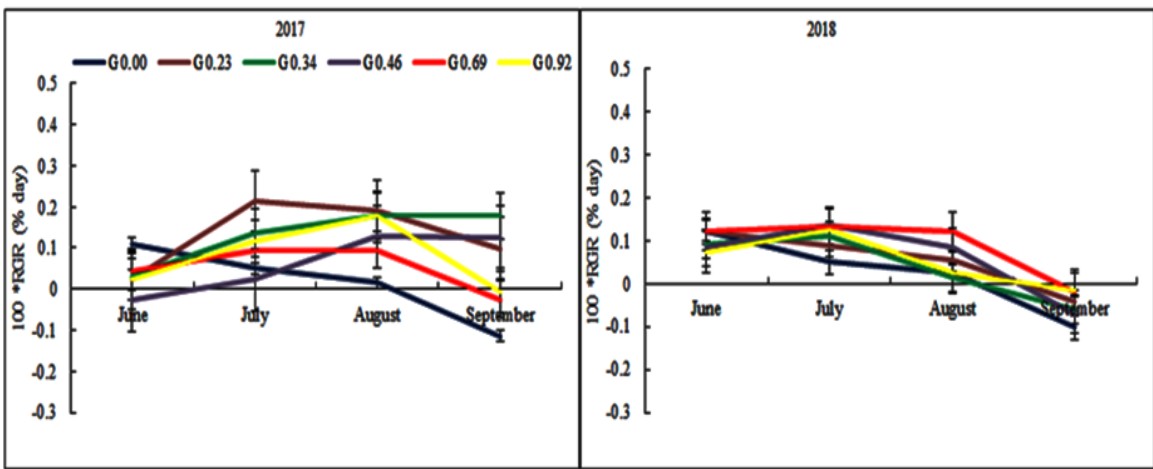

**Figure 7.** Effect of grazing intensity and season on relative growth rate (RGR).

*3.6. The Relationships of ANPP with Soil Moisture, Soil Temperature, and Soil Nutrient Contents*

Soil moisture had a positive relationship with ANPP ($R^2$ = 0.0748; Figure 8a), and the best relationship was between soil temperature and ANPP ($R^2$ = 0.3544; Figure 8b), which was a negative relationship. The relationship between ANPP and soil total nitrogen (TN) was also negative ($R^2$ = 0.0426; Figure 8c), while with total phosphorus (TP), a positive relationship was observed ($R^2$ = 0.1465; Figure 8d).

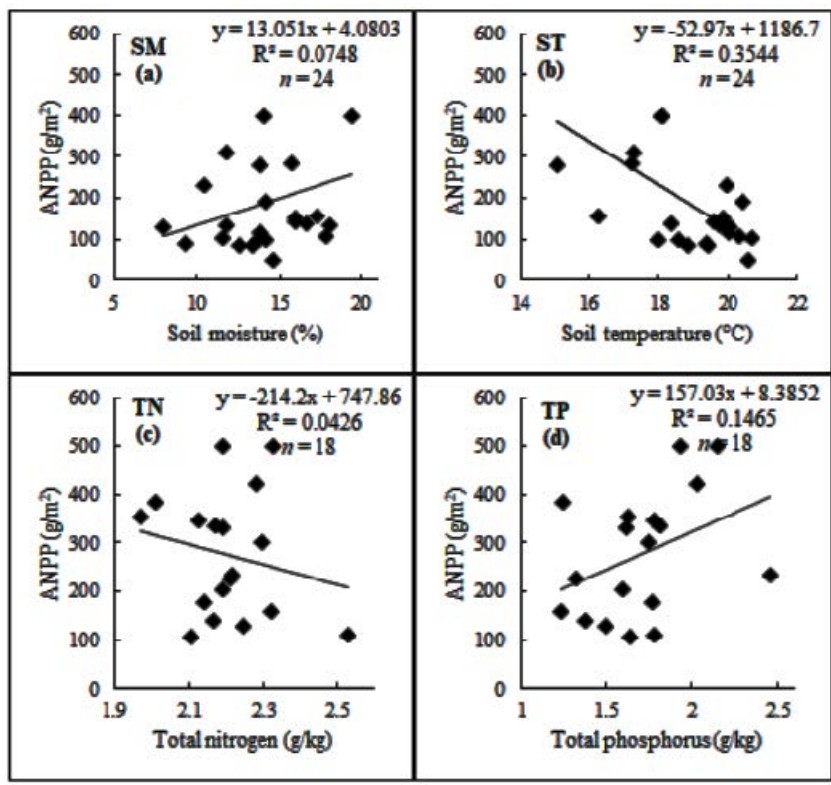

**Figure 8.** Relationships between ANPP and soil moisture (SM) (**a**), soil temperature (ST) (**b**), soil total nitrogen (TN) (**c**), and total phosphorus (TP) (**d**).

*3.7. The Relationship of Plant Components and Soil Factors*

The principal component analysis (PCA) was used to explain the relationship between the plant components and soil factors. The two axes of PCA can clarify the difference in the dataset (Figure 9).

C4 and C3 species showed a converse relationship. ANPP and belowground biomass (BGB) are naturally non-antagonistic. SM, ST, and soil bulk density (SBD) were also in converse relationships. The soil nutrient components, TN, and TP were more related to heavy grazing. Soil moisture was closely related to non-grazing, light grazing, and moderate grazing. C3 species were related closely to heavy grazing (G0.92), and C4 species were closely related to non-grazing and moderate grazing, whereas ANPP, BGB, and SM were closely related to non-grazing (G0.00), light grazing (G0.23), and moderate grazing (G0.46). Moreover, ST and SBD were closely related to the moderate grazing (G0.46) and heavy grazing (G0.92).

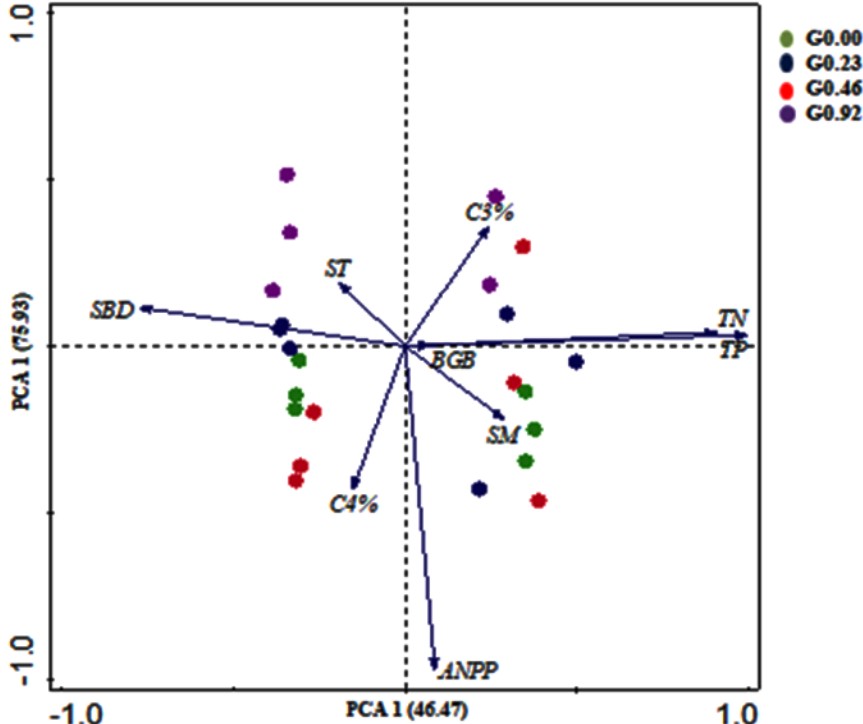

**Figure 9.** PCA ordination of 24 samples with five environmental variables and three biological variables in the Hulunber grassland, China. The vectors shown represent the major explanatory environmental variables. SBD, soil bulk density; SM, soil moisture; ST, soil temperature; TN, total nitrogen; TP, total phosphorus. The biological variables: BGB, belowground biomass; C4%, C4 species proportion; C3%, C3 species proportion. The circles represent plots.

## 4. Discussion

In 2018, the productivities of ANPP, AGB, and herbage growth were higher than in 2017, and this observation corresponded with the higher precipitation rates in June and July of 2018. Precipitation in June and July was 72.5 and 117.2 mm, respectively, in 2018, whereas in 2017, only 11.4 and 53.2 mm can be estimated at the same time (Figure 2). Several researchers have discussed the strong effect of rainfall and rainfall change on grassland yield in Inner Mongolia [8,9,13,47,48]. Both precipitation and variability are used to account for the equilibrium of the non-equilibrium systems [49,50]. Therefore, in arid and semiarid environments, when the annual rainfall is between 250 and 500 mm, and the annual rainfall change rate is more than 30%, the non-equilibrium system can be predicted to some extent. Some equilibrium features of grasslands have been found in this study—notably responses of vegetation to increasing grazing intensities, but also powerful impacts of inter-annual rainfall variability, an attribute of non-equilibrium systems. Therefore, both elements of equilibrium and non-equilibrium systems existed in the Inner Mongolian steppe system, which seems to be basically right on the semiarid grasslands [8,48,51–53]. The theory of non-equilibrium application was generally challenged [54], and it was concluded that, although the inter-annual variation of precipitation has

a strong short-term effect, the vegetation productivity decreases significantly with the increase of grazing. Moreover, the vegetation productivity substantially decreases with heavy grazing, in spite of a powerful short-term impact of inter-annual variation in precipitation. Restoration of grassland is a long-term process [55]; the short-term grazing effect may occasionally be reduced due to the change in rainfall.

### 4.1. Grazing Effects on Aboveground Biomass (AGB)

Grazing is one of the essential methods for humans to utilize grasslands. Grazing is also considered to be a complex disturbance path on grassland, which not only has affirmative effects on vegetation community, but also has an undesirable impact [56–58]. Due to the differences in grassland types, climate, and grazing history, the impacts of grazing on grassland are often diverse [59], but it is commonly believed that the rangeland community merits are strongly associated with grazing pressure [60,61]. The herbage mass is one of the important community merits, and the grazed grassland ecosystem stability can be reflected through herbage mass under different treatments of grazing, and it could also be used to measure the growth condition, succession tendency, production capability, and pasture's carrying capacity. It is a generally held view that there occurs an appropriate system of grazing management that can be used to attain a suitable utilization of the grazed grassland ecosystem [14]. Grassland productivity decline may be caused due to overgrazing [62] and the devastation of the grassland ecosystem functioning, which finally results in grassland degradation [63–65]. In the present study conducted at an Inner Mongolian typical steppe ecosystem, grazing reduced the grassland herbage quantity. Moreover, the AGB of the grazed treatments was significantly lower when compared to nongrazing treatments. The AGB significantly decreased with an increase in grazing intensity (Figure 3). Therefore, our results do not seem to support the hypothesis of grazing optimization that the plants of a grassland produce more biomass under moderate grazing than under non-grazing, due to overcompensation. The overcompensation relies on several factors, such as the grazing disturbance recurrence and magnitude, pattern of grazing, species, and kinds of animal [66,67]. Existence of under-compensation was found in the present study, which was carried out in a zone with a long history of grazing by domestic animals. Several studies showed that under-compensation predominantly happens in grasslands which have a long history of grazing [66,67].

### 4.2. The Effect of Grazing Intensity on Aboveground Net Primary Production (ANPP)

Previous studies have shown that the annual precipitation is positively correlated with ANPP, and the yield difference among different years can be explained by the change of annual rainfall first. This relationship is often recorded on semiarid grasslands [5,8,9,13,14,31] and seemed to be a common response of Inner Mongolia steppe ecosystems [68]. In contrast with the idea of grazing optimization, which was mentioned to be a conceivable response to grazing in steppe ecosystems [14,32,35,69], the consistency of the evidence of the induced optimization effect of grazing or clipping was not confirmed. Nevertheless, whether there are affirmative impacts [32,33,69], passive impacts [70,71], or neither [14,69] on the net initial production, the response of ANPP to grazing or defoliation remains controversial. Briske [66] clarified the difference in results and conclusions previously reported by authors and that they were drawn from their studies with various approaches, backgrounds, and grassland conditions. However, the universal patterns (including community level and species level) revealed the overall negative impact of grazing on ANPP in a semiarid grassland [72]. The results of this study showed that grazing reduced ANPP, which can be explained by the stress level of plants (Table 1 and Figure 4). This might be attributed to the damage of plant tissues resulting from leaves lost and treading [72] or to the lowering soil moisture [8,9]. Short-term grazing optimization indications have also been found at lightly (G0.23) grazed plots in 2017 and at zero (G0.00) grazed plots and lightly grazed (G0.23) plots in 2018. In addition, with the extension of grazing time, the adverse effect of grazing on ANPP is more and more obvious, thus highlighting the time value of the grazing effect. The ANPP of grassland is correlated directly to its capability of grazing, which is identified as the cattle

number per unit area that can be sustained for a certain period. Diminishing ANPP, thus, leads to decreasing economic benefits [9,73].

### 4.3. Grazing Effects on the Forage Dry Matter Intake

Increasing grazing intensity will reduce the quality of existing forage and the supply of DMI, which has been confirmed in some studies [5,8,74,75] and is also evident from current study. On highly intensively grazed grasslands with scattered vegetation cover, the inadequate or a complete lack of protection against sandstorms, particularly in winter and spring, promotes desertification and may reduce the productivity of grassland in the long-term [11]. In contrast, Reference [76] reported that heavy grazing maintains short grass steppe in a suitable state for high production. Furthermore, DMI on heavily grazed plots G0.69 and G0.92 decreased in this study, in September 2017, to a level where cattle survival may not be guaranteed anymore, and the animals had to be taken away from grasslands, indicating that the variation of the precipitation amount among years influences the effect of heavy grazing (Figure 5). Zoby and Holmes [77] and Langlands and Bennet [78] reported that a linear decline in DMI and DMI with increasing grazing intensity, as well as a close relationship between herbage mass and herbage intake of cattle grazing at high grazing intensity with low herbage mass, has been observed [77]. In contrast, on grassland with enough available herbage, no impact of an additional increase of herbage mass on feed intake was determined in this study. The present results were in line with reports by other authors [78,79] who reported an asymptotically increase in DMI. Moreover, DMI increased with increasing herbage mass, until it reached a maximum level where it stayed independent of additional increases in herbage mass. In our study, a considerable decline in herbage mass with increasing grazing intensity was observed (Figure 5). This appears to be a mutual response of herbage, especially in the short term, as observed in this steppe ecosystem, a phenomenon attributed to sward damage by grazing and trampling on the plants and thus unable to overcompensate [8,9]. However, it has been debated whether grazing at a given intensity level would increase grassland primary production [32,80] or whether it would normally decrease as a result of grazing.

### 4.4. The Effect of Grazing Intensity on the Herbage Growth

RGR is considered to be an important parameter to measure the response of plants to grazing. The yield per unit of production organization can be expressed in terms of RGR [34] Therefore, RGR is the correct answer to the question of whether grazing causes plant compensation response [72]. Reference [34] reported that compensatory growth is a result of any considerable increase in RGR due to grazing; however, it does not necessarily include increased forage accumulation. In our study, RGR shows different patterns between different years, highlighting the importance of water. In the wet year of 2018, compared with the dry year of 2017, RGR showed more significant compensation growth. The significant effect of grazing rate on RGR was reflected in the damage response of grazing under G0.92, as compared to G0.23. Therefore, this may be due to the high-stress load caused by dryness and overgrazing (Table 1). Our study outcomes agreed with the results of References [8,9,81,82], who mentioned that the severe clipping of plants revealed more compensatory growth during the moist years than dry years. In contrast to the passive response of RGR to the stocking rate (G0.92), in 2017, the RGR in the higher rainfall year (2018) recorded no negative effect by grazing even with the stocking rate of G0.92. To some extent, the RGR data observed in 2018 indicates that the response of compensation of plants to medium and high pressures of grazing is not substantial (Figure 7 and Table 1). The reverse growth model between years showed that the higher the grazing intensity, the higher the utilization efficiency of rainwater. The effect of rainwater utilization efficiency on light-grazing grassland has been previously discussed [81]. Compensatory growth is regarded as a mutual response of plants to grazing, which is affected and subjected by different mechanisms [72]. The distribution of carbohydrates, both already stored and recently produced, enhanced regrowth of leaves at the expense of roots and seems to be an essential factor in controlling responses of

compensatory growth [8,9,69,80–82]. Not only the carbohydrates redistribution, but also the nitrogen reserves mobilization from roots and stems for re-growing leaves, might positively impact plant regrowth [83–85]. The results of the grass absolute growth represented by GR showed that, with the passive response to G0.92 becoming obvious, the G0.23 supported the GR (Table 1), and a comparable pattern appeared between the years. The results of this study showed that compensatory growth is a joint response of grazing to precipitation changes over the years. However, the response of RGR and ANPP to grazing intensity showed that compensatory growth could not completely compensate the significant decrease of grazing cattle biomass (Table 1). No overcompensation was detected, so it can be said that the plant grows at its prospective maximum relative growth rate [8,9,34]. Nevertheless, compensatory growth alleviated grazing passive effects on ANPP.

### 4.5. Relationships of ANPP with Soil Moisture, Soil Temperature, and Soil Nutrient

The relationship of ANPP with soil moisture and soil temperature has always been a mystery on a global scale. Nevertheless, these relationships are important to understand the potential impact of climate change on primary production. SM is considered to be an important indicator of ecosystem response to climate change [86,87]. ANPP is positively correlated with SM (Figure 8a), indicating that SM can predict the change of ANPP better than precipitation [88]. Compared with precipitation, SM has a greater impact on ANPP, mainly due to the following two reasons; first, although precipitation in the growing season is considered to be a predictor of ANPP, soil water is directly related to root activity, plant water condition, and photosynthesis [89,90]. The availability of other soil resources also changed for a long time, through soil water dynamics [91]. Secondly, SM is mediated by the water-storing capacity of the soil, which better expresses the water availability for plant growth than rainfall [92]. A positive relationship of ANPP with SM and ST was observed in our study (Figure 8a,b). In fact, the major role of mean annual precipitation in arid alpine grassland and semiarid grassland has been confirmed by previous researches [93–96], but the contribution of temperature has also been confirmed by other research studies in Inner Mongolia grassland and tundra meadow [97,98]. In addition, previous studies have shown that precipitation is the first determinant of grassland, followed by temperature [94], which may also be applicable globally. Some studies have also reported a better correlation between rainfall and ANPP than temperature [99], which was also similar to our results.

In accordance with previous studies on this ecosystem [100–102], we found that ANPP was positively associated with soil nutrient contents (Figure 8c,d), and the productivity of the terrestrial ecosystem was generally considered to be limited [89]. Although nitrogen (N) is considered to be the key determinant [103] of ANPP, the prevalence of the common limitation of nitrogen and phosphorus (P) is increasingly recognized [104,105]. However, it is still unclear to what extent land productivity is limited by nutrients other than nitrogen and phosphorus. Many studies focused on the limitation of single-nutrient nitrogen [103]. The role of extra nutrients is increasingly recognized in grasslands and other systems. A recent meta-analysis of 1400 N and P fertilization studies [105] showed that both N and P restricted land productivity. In addition, these nutrients are usually co-limited, and their co-limited productivity exceeds the sum of their respective limits. This meta-analysis is by far the most comprehensive assessment of nutrient constraints in ecosystems. However, in natural systems [106,107], including grasslands [103], a critically endangered community [108], which accounts for about one-third of the earth's terrestrial net primary production, little is known about the global scope and extent of multiple constraints on nutrients, other than nitrogen and phosphorus. Moreover, the multiple nutrient restrictions have not been tested on grassland with standardized experimental methods all over the world. The lack of single- or multi-nutrient restriction could not be explained, which means that we may miscalculate the magnitude and extent of nutrient restriction of NPP. The synergetic restriction of N and P on ANPP of grassland at the global average level contradicts the viewpoint that N is the main nutrient [103] which has been considered to limit grassland productivity for a long time, and the key role of P is emphasized.

*4.6. Relationship of Plant Components and Soil Factors*

The PCA shows the grazing intensity effects size on environmental variables and grassland vegetation. Moderate grazing (G0.46) and heavy grazing (G0.92) are ascribed to SBD. Light grazing (G0.23), moderate grazing, and heavy grazing are associated with an increase in soil nutrient contents (TN and TP). Soil moisture (SM) is closely related to non-grazing (G0.00), light grazing, and moderate grazing, while the increase in soil bulk density is associated with non-grazing, moderate grazing, and heavy grazing. Moreover, non-grazing, light grazing, and moderate grazing are associated with an increase in ANPP, BGB, and soil moisture. C3 species were more related to heavy grazing, while C4 species were closely related to non-grazing and moderate grazing (Figure 9). The findings of our study implied that important references for assessing the scenarios of grazing management can be provided through monitoring changes in environmental variables, ANPP, BGB, C3 species, and C4 species in the communities of grassland and applying timely adaptive practices to maintain the ability of long-term grassland for sustainable productivity.

## 5. Conclusions

In this study, different sensitive indexes of short-term grazing were identified and evaluated. Grazing intensity had a negative effect on ANPP. However, compensatory growth is a mutual response to increased grazing intensity but seems to be controlled by rainfall. There was a significant negative linear correlation between the grazing intensity and the negative grazing effect on the aboveground biomass (AGB). Therefore, the grazing intensity gradient well reflects the grazing pressure from nongrazing to continuous grazing. The productivity of grassland is affected negatively by strong grazing, so AGB is regarded as a potential sensitive short-term index. In this study, irrespective of the management system used, G0.23 and G0.92 respectively represent the best and worst response of AGB to grazing under light and heavy grazing stress. This shows that the optimal grazing point was reached at G0.23 treatment. On the other hand, in the short term, especially in the high precipitation year of 2018, plants show a grazing response to the increase of grazing intensity. Understanding the ecological consequences of grazing intensity provides useful information for assessing current grazing management scenarios and taking timely adaptation measures to maintain grassland capacity in the long-term system.

**Author Contributions:** Conceptualization, A.I.A., X.X., and R.Y.; main analysis, A.I.A. and L.H.; visualization, A.I.A, Y.M.Z., and L.H.; writing original draft, A.I.A.; supervision, X.X. and R.Y.; funding and project administration, X.X.; writing–review and editing, X.X. and R.Y. All authors have read and agreed to the published version of the manuscript.

**Funding:** This study was supported by the National Key Research and Development Program of China (2016YFC0500608), the Fundamental Research Funds for Central Nonprofit Scientific Institution (1610132019031, 1610132019040, and Y2019YJ13), and Special Funding for Modern Agricultural Technology Systems from the Chinese Ministry of Agriculture (CARS-34).

**Acknowledgments:** The authors gratefully acknowledge the financial support from the Chinese Government Scholarships (CGS) and the Chinese Academy of Agricultural Sciences, Institute of Agricultural Resources and Regional Planning. We are grateful to many colleagues at the Institute of Agricultural Resources and Regional Planning and to all staff of the Hulunber Grassland Ecosystem Research Station for assistance with field observations and sample collection.

**Conflicts of Interest:** The authors declare that they have no conflict of interest.

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
