# Peer review of "The Joint Effect of Grazing Intensity and Soil Factors on Aboveground Net Primary Production in Hulunber Grasslands Meadow Steppe"

_agriculture, doi:10.3390/agriculture10070263_

Round 1

Reviewer 1 Report

Dear Authors,

General

The manuscript adds good literature in the field of grazing lands. Overall, the experiment was conducted well, and good data was collected, however, the writing and English structure require significant improvement.

There are plenty of grammatical and proofreading errors. This manuscript needs a thorough revision of grammar and sentence structures.

Abstract

Line 13: This sentence is vague. What do you mean by various practices? Please simplify.

Line 15: Simplify the sentence by listing objectives with (i) and (ii)

Line 19: Abbreviations need to be expanded at the first appearance. I think AU was mentioned here first. Why is there a period (.) between AU and ha-1? Also, make sure the formatting of units is correct throughout the manuscript.

Line 23: AGB needs to be expanded. Please expand all abbreviations at their first appearance here and throughout the manuscript.

Line 25: Effect of rainfall is highlighted in the abstract, however, rainfall is not mentioned in the methods and the title? Why are you putting so much emphasis on rainfall when it was not a factor under consideration?

Introduction

The introduction is not thorough and needs significant improvement before publication. For example;

Soil factors are mentioned in the title, however, there is no mention of soil factors in the introduction. A thorough literature review of soil factors affecting grazing lands is required.

Line 54-56: Not clear. Simplify.

Line 59: This sentence is not grammatically correct. Replace by “Inner Mongolia lack detailed information”. Please fix the sentence structure and grammar here and throughout.

Line 48-58. This section of the introduction needs to be expanded to highlight past studies and their results. This will strengthen your objectives.

Materials and Methods

Line 82. I think the elevation needs to be masl not just m.

Lines 80-82 and Lines 93-95 are exactly the same. Please remove the redundant part.

Line 97: Fix the “where1Anumal Unit”. Please conduct a thorough proofreading before the next submission.

Line 109: Expand DMI

Line 123: Space between 29] and the period (.) Please check this throughout.

Line 149-54. Add references to each soil testing method and briefly describe it.

Line 156-157: Fix the sentence structure. Readability is low.

Line 158: In order of testing the interaction? What does this mean? Fix sentence structure, here and throughout.

Results

The results section is written and organized well.

Line 165: “was far away” is not a good term. Replace by “lower than the long-term average”.

Figure 5: The figure resolution could be better. Maybe splitting figure 5 into two figures might improve readability (one for GR and one for RGR).

Line 241: Please introduce C3 and C4 plants in the Methods section.

Discussion

Line 288: You can add some references here to highlight grassland degradation. Please try to add new studies that are pertinent to your study. For example:(i) https://doi.org/10.2489/jswc.65.3.200  (ii) https://doi.org/10.2489/jswc.74.4.323 , (iii) https://doi.org/10.1002/agj2.20189 

Line 364-366: Recent references would help this argument. I suggest adding (i) https://doi.org/10.1016/j.agsy.2020.102847 (ii) https://doi.org/10.3390/su12020558 among others, which show an increase in mineralization of N due to increased grazing.

Line 380: fig.6a. Please be consistent with formatting. Change by “Fig. 6a.”

Line 400: Space after [87].

Line 404. Why is “Co Limited” capitalized?

Conclusion

Clear and succinct.

References

Formatted well.

Author Response

Dear reviewers: 

Thanks very much for your careful review and constructive suggestions with regards to our manuscript"The Joint Effect of Grazing Intensity and Soil Factors on Aboveground Net Primary Production in Hulunber Grasslands Meadow Steppe". Those comments are helpful for the author to revise and improve our paper. We have studied the comments carefully and tried our best to revise and improve our manuscript and made great changes in the manuscript according to the referees' good comments. Revised portion is marked in red in the paper. We appreciated for Editors/Reviewers warm earnestly, and hope that the corrections will meet with approval. Please feel free to contact us with any question and we are looking forward to your consideration.

Best regards,

(ID: agriculture-8564483)

Reviewer 2 Report

I enjoyed reading this article, but the writing needs improvement before publication. I have provided some suggestions in sticky notes in the attachment. Also, I urge the authors to peruse literature outside of their region and contrast or compare their findings. This addition would help improve applicability outside their study area.

Author Response

Dear editors and reviewers:

Thank you very much for your careful review and constructive suggestions with regard to our manuscript “The Joint Effect of Grazing Intensity and Soil Factors on Aboveground Net Primary Production in Hulunber Grasslands Meadow Steppe” (ID: agriculture-8564483). Those comments are helpful for authors to revise and improve our paper. We have studied comments carefully and tried our best to revise and improve the manuscript and made great changes in the manuscript according to the referees’ good comments. The revised portion is marked in red in the paper. We appreciate for Editors/Reviewers’ warm work earnestly and hope that the corrections will meet with approval. Please feel free to contact us with any questions and we are looking forward to your consideration.

Best regards,

(ID: agriculture-8564483) authors.

Reviewer 3 Report

This paper has a good base, but it really needs to be looked over very closely for grammar.

These types of projects have been conducted for 50+ years all over the world; however, it may be of interest to land managers in Inner Mongolia, where the paper takes place. That alone I think makes it worthy for eventual publication, as different ecotypes respond differently to grazing. 

The authors had a continuous grazing period that lasted from June to September and found that C3 plants thrived and C4 plants were hindered. This should be expected, as C3 grasses are cool season (fall-late spring) plants that would not be under as much grazing pressure with a summer grazing system.

OMI is not mentioned in the paper until line 330, and it is not a defined term; yet it is reported in the results. Please define this term and how it was measured.

Paper needs extensive review of grammar used throughout.

Line 33. Insert “a” between “forms” and “portion”

Line 36. Change “have been transformed” to “has been transformed”

Line 37. Hyphenate “semi natural” (semi-natural)

Line 51: change “plant” to “plants” and insert “a” before “grazing system”

Line 56: “A little experiments of controlled grazing….” Does not sound right, consider re-wording

Line 59: change “lack” to “lacking”

Line 62: insert “an” after “to find”

Line 65: change “needful” to “needed”

Lines 68-72: run-on sentence, consider re-wording into shorter sentences

Line 80: consider adding a map of the study location for those that are not familiar with the region

Line 88: Scientific names are lacking authority

Line 88: the word “and” after scorzonerifolium should not be italicized

Lines 93-95: this sentence is a copy/paste from the previous section, please remove

Line 95: change “The experiment of grazing…” to “The grazing experiment….”

Line 97: add a space at the end of the sentence “…adult cows).”

Line 98: change “covering” to “covered”

Line 100. Delete space between “plot” and the period “.”

Line 100: change “holded” to “was held”

Line 101: insert “the” before “2017”

Lines 103-104: bad grammar. Suggest rewording to “Measurements were made three times per month in 2017, and four times per month in 2018; excluding September where 3 measurements were taken due to prevailing weather conditions.”

Line 106: replace “growth” with “growing”

Line 114: remove “dried on an oven-dry at” and replace with “oven-dried for”

Line 115: remove space between [27] and “.”

Line 115: insert “were” between “materials” and “weighed”

Line 123: insert “the” between “by” and “moving”

Line 123. Delete space between [28,29] and the period “.”

Line 125. Delete space between “September” and the comma “,”

Line 126: remove the exponent “2” from “50x50cm”

Line 140: remove space between “un-“ and “grazed”

Line 142: insert “were” between “USA)” and “used”

Line 150: insert “of” after “analysis”

Line 150: insert “a” between “by” and “Mastersizer”

Line 156: insert “analysis” after “The data”

Line 156: insert a space between “Program” and “(SAS 9.2)”

Lines 156-157: delete “the analysis included”

Line 157: change “exam” to “examine”

Line 157: there is a double space before “One way ANOVA” change to a single space

Line 157: hyphenate “One way ANOVA” (One-way ANOVA)

Line 158: change “of testing” to “to test”

Line 159: add a comma “,” after “and year”

Line 165: change “2107” to “2017”

Line 165: replace “far away from” to “well below”

Lines 166 and 167: change “the average of long term” to “the long-term average”

Line 170: suggest changing “epic” to “peak”

Line 176: change “(AGB) is higher” to “(AGB) was higher”

Line 189: change “with the increase of grazing intensity” to “with increased grazing intensity”

Line 191: capitalize the “g” on “g0.23”

Line 193: capitalize the “g” on “g0.92”

Line 195: change “mid of August” to “mid-August”

Line 198: I just want you to double check figure 3 ANPP 2018, G0.92. it is so much lower than all of the other values, but is marked with “b.” I just wanted to make sure that was correct and did not need a “c”

Line 217: “with a further declining registered constantly” makes no sense to me, I am not sure what is being said here. Do you mean “with a continued decline to the lower level?” whatever the meaning, I suggest re-wording this passage.

Line 223. Add a space after “2017”

Line 241: add “a” between “showed” and “converse”

Line 245: add a space before “C3”

Line 245: your continuous grazing season was from June to September, so the fact that that favors C3 plants is no surprise. C3 grasses are cool season grasses, and so they will thrive and reproduce since all of the grazing occurred in the warm season, which would be to the detriment of C4 grasses.

Line 253. Space is on the wrong side of the period “.” Between “variables” and “SBD”

Line 254. Space is on the wrong side of the period “.” Between “phosphorus” and “The”

Line 255: space is on the wrong side of the period “.” Between “proportion” and “And.”

Line 255: Remove “And” from the start of a sentence. Re-word to “The circles represent plots”

Line 260: change “researches” to “researchers”

Line 273: change double space to single space between “in spite of” and “a powerful”

Line 277: change “human” to “humans”

Line 284: “pastures” in this usage is possessive, so I suggest changing to “pastures’” with an apostrophe

Line 287: change “devastating” to “devastation”

Line 295: “existed” doesn’t really fit here. Perhaps “Existence of”

Line 308: Starting a sentence with a citation [52] is somewhat confusing, especially on a line break. I will defer to the editors on this one, but I feel that when making a direct reference to someone, you should write is such: “Briske [52] clarified…”

Line 311. Hyphenate “semiarid” (semi-arid)

Line 314: change double space to single space between “lowering” and “soil”

Line 325: insert “and” after “desertification”

Line 326: change “on the long-term” to “in the long-term”4

Line: 329: again, starting a sentence with a citation number. I suggest “Zoby and Holmes [62] and Langlands and Bennet [63]….”

Line 330: this is the first mention of “OMI” in the paper. You need to go back and define this term, and if you measured it, it needs to be in the methods section.

Line 335: change double space to a single space before “Moreover”

Line 380: capitalize “fig. 6a”

Line 388: capitalize “fig. 6a”

Line 396: capitalize “fig. 6c”

Line 399. Fix space/period “.” Before “However”

Line 401: Fix space/period “.” Before “The role of extra”

Line 404: hyphenate “co limited” (co-limited)

Line 409: change “nutrition” to “nutrient”

Line 422: remove space after “non-“

Line 423: change “C3 species was more” to “C3 species were more”

Line 424: Change “C4 species was more” to “C4 species were more”

Line 435: is “re grazing” supposed to be “re-grazing?” or possibly “continuous grazing?” I’m not certain as to what is being said here.

Lines 458-637: don’t use “et al” in the references list, spell out all authors.

Lines 458-637: add DOIs to references where applicable

Line 463; 480; 488; 492; 494; 513; 519; 520; 548; 572; 596; 602; 603; 614; 624; 628; 630; 636: Capitalize Journal names in references

Author Response

(The authors gave the same response as above.)

Round 2

Reviewer 1 Report

Dear Authors,

Thank you for your extensive revision. My concerns have been addressed. I recommend publication of this manuscript.

Thank you.

Author Response

Dear editors and reviewers:

First of all, we want to express our thanks for your very rapid response to our revised paper! “The Joint Effect of Grazing Intensity and Soil Factors on Aboveground Net Primary Production in Hulunber Grasslands Meadow Steppe” (ID: agriculture-8564483). We are very pleased with your appreciation to our effort. Thanks very much for your recommending our paper for publication. We appreciate for Editors/Reviewers’ warm work earnestly, and the huge effort made by you during all process regarding our paper. Please feel free to contact us with any questions and we are looking forward to your consideration.

Best regards,

Authors

Reviewer 3 Report

First of all, I want to express my thanks for your very rapid response to the review comments! I am very please with the turnaround and quality of the edits made to my, and the other reviewer's suggestions.

I have just a few very minor grammar edits for the new version, otherwise I am satisfied with the current state of the paper.

Line 49: make "ecosystem" plural, "ecosystems"

Line 81: add "of" after "wide-range" so that it will read "This wide-range of stocking rates"

Lines 68 and 86: change "above-ground" to "aboveground" to match the spelling used throughout the rest of the paper

Cheers!

Author Response

Dear editors and reviewers:

First of all, we want to express our thanks for your very rapid response to our revised paper! The Joint Effect of Grazing Intensity and Soil Factors on Aboveground Net Primary Production in Hulunber Grasslands Meadow Steppe” (ID: agriculture-8564483). We are very pleased with your appreciation to our effort and we are so happy for your recommending our paper for publication. “We have studied comments carefully and revised all mentioned minor revision. The revised portion is marked in blue in the paper. We appreciate for Editors/Reviewers’ warm work earnestly and hope that the corrections will meet with approval. Please feel free to contact us with any questions and we are looking forward to your consideration.

Best regards,

Authors
